# Reducing Methane Emissions with Humic Acid–Iron Complex in Rice Cultivation: Impact on Greenhouse Gas Emissions and Rice Yield

Hyoung-Seok Lee [1,2], Hyo-Suk Gwon [1], Sun-Il Lee [1], Hye-Ran Park [1,2], Jong-Mun Lee [1], Do-Gyun Park [1,2], So-Ra Lee [1], So-Hyeon Eom [1] and Taek-Keun Oh [2,*]

1   National Institute of Agricultural Sciences, Rural Development Administration, Wanju 55365, Republic of Korea; lhs0218@korea.kr (H.-S.L.); gwonhs@korea.kr (H.-S.G.); silee83@korea.kr (S.-I.L.); parkhr098@korea.kr (H.-R.P.); jmlee1019@korea.kr (J.-M.L.); sacred86@korea.kr (D.-G.P.); dlthfk789@korea.kr (S.-R.L.); sh3007@korea.kr (S.-H.E.)
2   Department of Bio-Environmental Chemistry, Agriculture and Life Sciences, Chungnam National University, Daejeon 34134, Republic of Korea
*   Correspondence: ok5382@cnu.ac.kr; Tel.: +82-428216735; Fax: +82-632383823

**Abstract:** Methane emissions from flooded rice paddies are a major source of atmospheric methane and represent a significant greenhouse gas with high climate-forcing potential due to anthropogenic activities globally. For sustainable agriculture, it is necessary to find effective methods for mitigating greenhouse gas emissions without reducing crop productivity. We investigated mechanisms to reduce methane emissions during rice cultivation by applying rice straw, rice husk biochar, humic acid, and a humic acid–iron complex, assessing greenhouse gases and rice yield over a single season. The results demonstrated that the treatment plots with rice straw and the humic acid–iron complex significantly reduced methane emissions ($563 \pm 113.9$ kg ha$^{-1}$) by 34.4% compared to plots treated with rice straw alone ($859 \pm 126.4$ kg ha$^{-1}$). Rice yield was not compromised compared to the control group treated with only NPK fertilizer, and growth in terms of plant height and tiller number was enhanced in the plots treated with rice straw and the humic acid–iron complex. Conversely, the plots treated solely with rice husk biochar and humic acid did not show a methane reduction effect when compared to the NPK treatment. The humic acid–iron complex has demonstrated potential as a methane mitigation agent with practical applicability in the field, warranting further long-term studies to validate its effectiveness.

**Keywords:** rice paddy; greenhouse gas; humic acid–iron complex; sustainable agriculture

## 1. Introduction

At the 26th Conference of the Parties (COP 26) in Glasgow in 2021, the international community announced the global methane pledge, aiming to reduce methane emissions by at least 30% by 2030 compared to 2020 levels [1]. Methane is considered the second most significant greenhouse gas in terms of anthropogenic climate forcing after carbon dioxide, with an estimated annual global emission of approximately 576 Tg [2]. Methane has a global warming potential approximately 15 to 34 times greater than carbon dioxide [3], and agriculture accounts for ~40% of total global methane emissions [4]. The primary greenhouse gases emitted from rice paddies are methane and nitrous oxide [5], with 80–90% of the methane emitted through the aerenchyma of rice during the growing season [6]. Methane emissions in rice cultivation are significantly influenced by water management methods, the amount and type of organic matter used as substrate for methanogens, and factors such as crop variety and soil texture [7–9]. Notably, water management has been identified as the most effective method for reducing methane emissions, with studies showing reductions of 52–55% with intermittent irrigation compared to continuous flooding

by Yagi and Minami (1990) and approximately 50% reduction with a single drainage according to Sass et al. (1992) [7,10,11].

Biochar is defined as a solid material produced by pyrolyzing biomass materials such as rice husks or wood under oxygen-limited conditions at temperatures above 350 °C [12]. The IPCC (2019) categorizes biochar according to the feedstock and pyrolysis temperature and assesses its potential for soil carbon sequestration when applied to agricultural lands. A meta-analysis centered on East Asia by Lee et al. (2023) indicated that biochar application in paddies can reduce methane emissions by up to 22.9% and increase both soil organic carbon (SOC) content and crop yield [13]. Biochar is known to enhance soil aeration and adsorption capacity due to its porous surface, thereby promoting methane oxidation [14].

Humic substances are redox-active and can serve as terminal electron acceptors in anaerobic microbial respiration [15], playing a crucial role in suppressing methane emissions in anoxic paddy soils. Such humic substances, serving as electron acceptors in anaerobic methane oxidation processes, are pivotal in mitigating methane emissions, with this principle applicable to other electron acceptors like iron or sulfate [16]. When electron acceptors are supplied to flooded paddies, their reduction and the concurrent loss of electrons by the paddy enhance soil oxidation, thereby suppressing methane-producing microbial activity [16,17]. Humic acids, characteristics of natural organic matter (NOM), strongly complex with trivalent iron, can rapidly reduce due to the humic acid–iron complex, potentially serving as a source of oxidative capacity and suggesting their utility as methane mitigation agents [18,19].

In South Korea, agriculture also represents a significant portion of methane emissions, with 42.8% of the total 27.3 million tons of $CO_2$-eq methane emissions in 2021 attributed to this sector, including 5.4 million tons from rice cultivation [20]. Consequently, South Korea has joined the international methane reduction pledge and legislated carbon neutrality, setting a target to reduce greenhouse gas emissions by 27.1% in the agriculture, forestry, and fisheries sectors by 2030 compared to 2018 levels (as of October 2021). South Korea has conducted studies to develop unique national coefficients for accurate greenhouse gas quantification, evaluating methane reduction techniques in rice paddies, such as mid-season drainage that has proven effective [20]. Further research into mitigation strategies has included the exploration of minimal tillage [21] and deep placement fertilization [22], the development of the methane-reducing rice variety Milyang 360, which reduces methane emissions by 16% [23], and the application of biochar [13,24]. We aim to discover new methane reduction methods by combining natural materials like humic acids and iron, and conducting field tests to evaluate greenhouse gas emissions when biochar produced from rice husks and straw is applied, thus assessing the potential and field applicability of these materials as methane mitigation agents.

## 2. Materials and Methods

### 2.1. Field Design and Soil Properties

The experimental site was located at the rice experimental fields of the National Institute of Agricultural Sciences, located at 166 Nongsaengmyeong-ro, Iseo-myeon, Wanju-gun, Jeollabuk-do, Republic of Korea (35°49′29.4″ N 127°02′36.6″ E), and conducted in 2022. Six treatment plots were established with each treatment field designed to facilitate three replicates. The main treatments involved the addition of organic and inorganic materials, and chemical fertilizers N-$P_2O_5$-$K_2O$ were applied as a base to all plots. The control group was treated with only chemical fertilizer (NPK), whereas the treatment groups included rice straw (ST), humic acid (HA), a combination of rice straw and humic acid (ST+HA), rice straw combined with a humic acid–iron complex (ST+HA-Fe), and biochar (Biochar) (Figure 1).

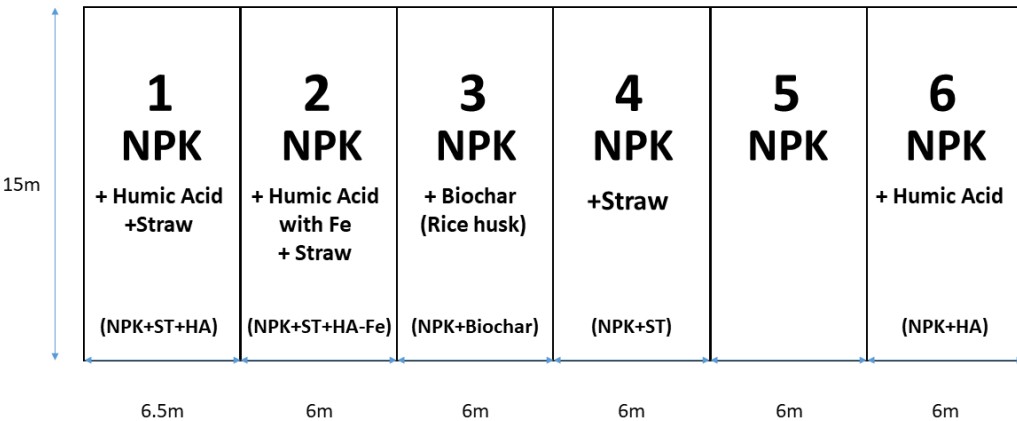

**Figure 1.** Treatment plots in rice paddy fields. Notes: NPK, application of N-P$_2$O$_5$-K$_2$O fertilizer; ST, application of straw; HA, application of humic acid; HA-Fe, application of humic acid–iron complex; Biochar, application of biochar.

### 2.2. Field Management and Materials Application

The rice variety used was *Oryza sativa* L. (Shin Dongjin), which was mechanically transplanted on 20 May with young seedlings at a planting distance of 30 × 20 cm. Water management for the paddy was uniformly maintained as continuous flooding for all six treatment groups, with drainage occurring one month before rice harvest. Fertilization of all treatment groups was performed according to the Rural Development Administration's fertilization standards for different crops [25]. For chemical fertilizer treatment (NPK), N-P$_2$O$_5$-K$_2$O was applied at a rate of 9-4.5-5.7 kg per 10 ares (Table 1).

**Table 1.** Schedule of rice cultivation.

| Irrigation | Basal Fertilizer and Application | Transplanting | Tillering Fertilizer | Heading Fertilizer | Drainage | Harvest |
|---|---|---|---|---|---|---|
| 13 May | 29 May | 30 May | 14 June | 8 August | 13 September | 11 October |

ST with a carbon content of 37.9% and a nitrogen content of 0.6% was applied at 600 kg per 10 ares. HA and HA-Fe were applied at 15 kg per 10 ares, based on the recommended fertilization rate for silicate fertilizers. HA was 94% pure (with 8% Potash) and produced by MYCSA AG (USA). HA-Fe was prepared by adding 0.5 kg of FeCl$_3$ [Iron (III) chloride anhydrous] to 2 kg of humic acid in 500 mL of distilled water with a pH of 10 and a temperature of 80 °C [26]. Biochar was produced using rice husks, and applied at 300 kg per 10 ares. Biochar's safety during the production was evaluated through component analysis. The components (N, C, H, S%) were analyzed using a Vario MACRO cube (ELEMENTAR, Langenselbold, Germany), and the oxygen content (O%) was analyzed using a Flash 2000 (Thermo Fisher, Monza, Italy) (Table 2).

**Table 2.** Elemental compositions of biochar.

| N | C | H | S | O | H/C | O/C |
|---|---|---|---|---|---|---|
| | | (%) | | | Ratio | |
| 0.82 | 56.27 | 0.96 | 0.54 | 3.88 | 0.21 | 0.05 |

### 2.3. Method of Gas Sampling and Analysis

Over 80–90% of the methane emitted from rice cultivated in paddies is released through the aerenchyma tissues [6]. Taking this into consideration, the closed chamber method was used to measure methane emissions. The chambers were made of transparent PVC arranged in a rectangular shape and were designed to open upward (Figure 2).

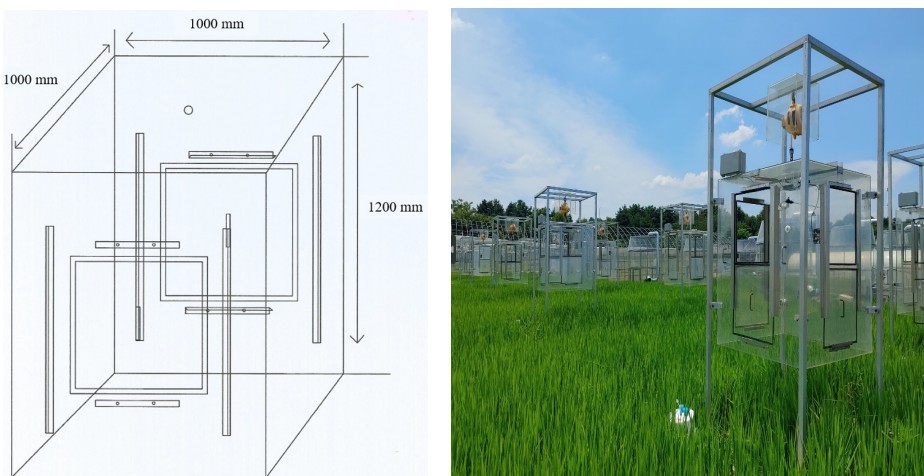

**Figure 2.** Specifications of bottom-up closed chamber and installation in rice paddy fields.

A fan was installed inside the chamber to circulate air for one minute before gas sampling to ensure the uniformity of air composition. A thermometer insertion point for measuring temperature was placed on the side of the chamber, alongside the sample collection port. The bottom of the chamber was designed to be filled with water to ensure airtight sealing when closed, while the top part could be lifted to allow ventilation and avoid interfering with rice growth when not in use for sampling. Sample collection was done between 10 a.m. and 12 a.m., considering the average daily peak hours of greenhouse gas emission during the rice cultivation period [27]. Greenhouse gas samples were collected weekly after rice transplantation by initially taking a 60 mL syringe sample with the chamber open, then closing and sealing the chamber to measure the internal temperature and volume. The fan circulated air within the sealed chamber, and a final sample was taken with a 60 mL syringe 30 min after sealing. The chamber was then opened until the next sampling event. The gas samples collected with the 60 mL syringes were quantitatively analyzed using gas chromatography (Gas chromatography: Detector-FID for $CH_4$, $CO_2$ and ECD for $N_2O$).

### 2.4. Method of Gas Flux and Greenhouse Gas Intensity Calculation

The gas flux per unit time and area, as well as the cumulative greenhouse gas emissions during the growing season, were calculated by Equations (1) and (2) below as previously described [10,21,28,29].

$$F = \rho \times V/A \times \Delta c/\Delta t \times 273/T \tag{1}$$

F = $CH_4$ or $CO_2$ or $N_2O$ flux (mg m$^{-2}$ h$^{-1}$)
$\rho$ = gas density (mg m$^{-1}$)
V = volume of chamber (m$^3$)
A = surface area of chamber (m$^2$)
$\Delta c/\Delta t$ = rate of increase in gas concentration ($\mu$L L$^{-1}$ h$^{-1}$)
T = absolute temperature (273 + mean temperature in chamber)

$$\text{Total } CH_4 \text{ or } CO_2 \text{ or } N_2O \text{ flux} = \sum_{i}^{n} (F_i \times D_i) \tag{2}$$

$F_i$ = the rate of flux (g m$^{-2}$ d$^{-1}$) in the *i*th sampling interval
$D_i$ = the number of days in the *i*th sampling interval
n = the number of sampling intervals

Methane emissions from the rice paddy were converted to $CO_2$ equivalents by Equation (3), and the net GWP was calculated [5,30,31].

$$\text{Net GWP} \left( \text{kg } CO_2 - \text{eqv.ha}^{-1} \right) = 21 \times CH_4 + 310 \times N_2O \tag{3}$$

The greenhouse gas intensity (GHGI), which represents the amount of greenhouse gas emitted per unit of yield, was calculated using the previously determined GWP as shown in Equation (4) [32,33].

$$\text{GHGI} \left( \text{kg } CO_2 - \text{eqv.kg}^{-1}\text{grain} \right) = \text{Net GWP}/\text{grain yield} \tag{4}$$

### 2.5. Soil and Crop Yield

An ORP-30-1-D platinum electrode (SWAT instruments, Dutch, The Netherlands) was installed in the soil to measure its oxidation-reduction potential (Eh value) in real-time. Simultaneously, a 5TM sensor thermometer (Campbell Scientific, Logan, UT, USA) was installed to measure the soil temperature. Samples were collected from the topsoil within 0–20 cm of the surface, and the collected soil was air-dried and sieved through a 2 mm mesh. The chemical characteristics were analyzed based on the soil chemical analysis methods of the Rural Development Administration [34]. The main soil characteristics were pH 5.59, with an organic matter content of 17.3 g/kg (Table 3).

**Table 3.** Chemical properties of soil before experiment.

| pH (1:5, $H_2O$) | EC (dS m$^{-1}$) | Av. $P_2O_5$ (mg kg$^{-1}$) | OM (g kg$^{-1}$) | TOC (g kg$^{-1}$) | T-N (g kg$^{-1}$) | Ex. Cation (cmol$_c$ kg$^{-1}$) | | |
|---|---|---|---|---|---|---|---|---|
| | | | | | | K | Ca | Mg |
| 6.1 | 0.3 | 25.9 | 17.3 | 10.0 | 0.7 | 0.35 | 4.4 | 1.68 |

Note: OM, organic matter; TOC, total organic carbon; T-N, total nitrogen; soil analysis method follows [34].

To identify factors that may influence methane production, an analysis of soil organic carbon was done to identify the carbon consumption pathways of methanogenic bacteria. Soil samples were collected 10 times from 27 January 2022 to 25 November 2022, and analyzed with the microbial-based hot water extraction carbon (HWEC) and water soluble carbon (WSC) methods. For WSC analysis, 3 g of soil (dry basis) was mixed with 30 mL of distilled water and stirred for 30 min, followed by leaching in a 20 °C water bath. The leachate was centrifuged at 3500 rpm for 30 min, and the supernatant was filtered through a 0.45 µm membrane filter. The filtrate was then measured with a TOC analyzer (Vario TOC cube, ELEMENTAR, Langenselbold, Germany). For HWEC analysis, 4 g of soil (dry basis) was mixed with 30 mL of distilled water and shaken for 10 s, then leached in an 80 °C water bath for 16 h. The leachate was centrifuged at 3500 rpm for 20 min, and the supernatant was filtered through a 0.45 µm membrane filter, followed by measurement with a TOC analyzer. At harvest, growth and yield surveys were conducted for each treatment group. The growth survey measured tiller numbers and plant height, with 20 plants per treatment group surveyed (on 11 October 2022). Crop yield was determined by harvesting 100 plants per treatment group on the same date as the growth survey, measuring grain moisture content, and adjusting the weight to a moisture content of 15% to calculate yield per hectare. The converted crop yield was used for the GHGI calculation.

### 2.6. Statistical Analysis

For statistical analysis, SPSS software (Version 26, IBM, Armonk, NY, USA) was applied. There were three replicates for each treatment group. To compare differences between independent treatment groups (Crop yield, $CH_4$ emissions, $CO_2$ emissions, HWEC and WSC Content), the Tukey test was conducted, and the significance level was set at 5%.

## 3. Results

### 3.1. Meteorological and Soil Conditions

During the rice cultivation period, the daily average temperature increased until July, reaching a peak temperature of 30.7 °C, followed by a gradual decrease from August. The total precipitation during the cultivation period was 733 mm, which was approximately 22% less than the same period of the previous year (937 mm) (Figure 3).

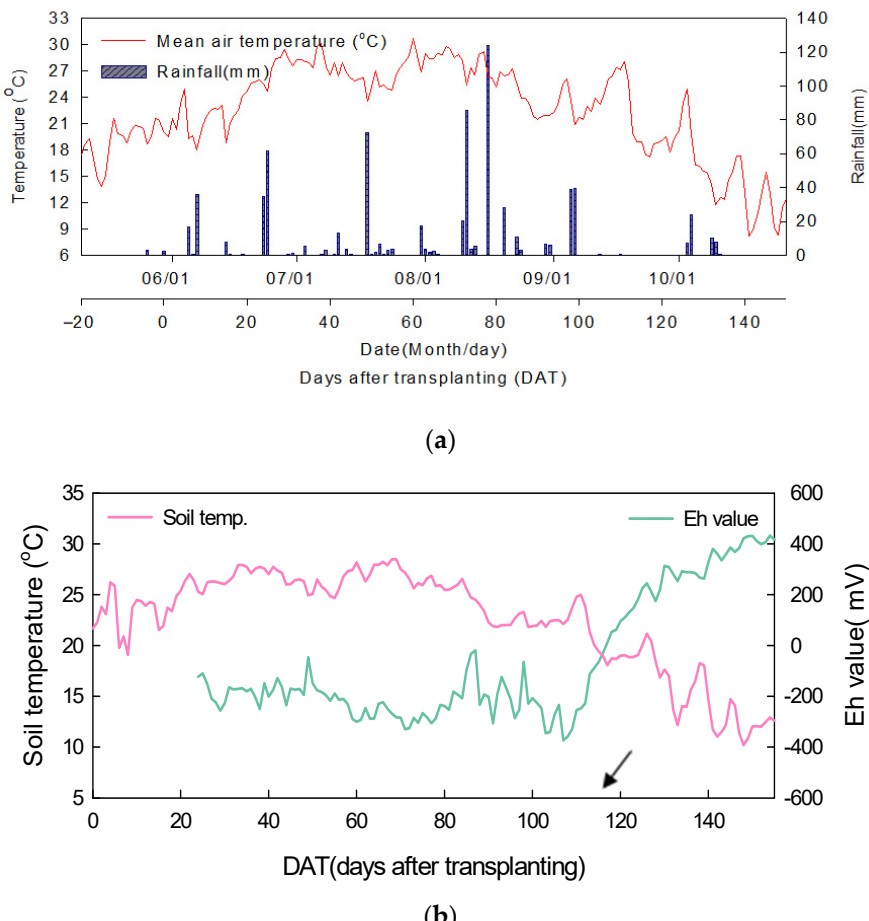

**(a)**

**(b)**

**Figure 3.** (**a**) Meteorological data; (**b**) changes in soil temperature and redox potential (↙: Complete water drainage).

Soil temperature fluctuations mirrored the daily average air temperature, showing a gradual decrease 80 days after transplantation. The optimal temperature range for methane production is 20 °C–40 °C, and the soil temperature dropped below 20 °C one month before harvest after complete drainage. The soil redox potential exhibited values around −200 mV because of the absence of midseason drainage, indicating a positive effect on the activity of methanogenic bacteria. However, after the implementation of complete drainage, the soil environment shifted to aerobic conditions, causing the redox potential to gradually increase and reach values of approximately 300 mV at the time of harvest.

### 3.2. Analysis of Soil Organic Carbon

The year-over-year changes in WSC (mg/L) and HWEC (mg/L) revealed no statistically significant differences between the treatment groups ($p > 0.05$). Specifically, the total WSC values for NPK, Biochar, HA, ST, ST+HA, and ST+HA-Fe treatment groups were 731.7 ± 82.36 mg/L, 800.6 ± 67.57 mg/L, 752.1 ± 16.84 mg/L, 786.6 ± 86.22 mg/L, 727.3 ± 36.85 mg/L, and 737.7 ± 73.91 mg/L, respectively, with no significant differences among them ($p = 0.745$). For total HWEC values, NPK, Biochar, HA, ST, ST+HA, and

ST+HA-Fe treatment groups were 812.3 ± 20.96 mg/L, 836.9 ± 20.84 mg/L, 822.1 ± 28.09 mg/L, 812.3 ± 41.84 mg/L, 826.0 ± 66.01 mg/L, and 800.4 ± 14.07 mg/L, respectively, with no significant differences ($p = 0.862$). Unlike the WSC values, however, the HWEC values measured after harvest day were relatively higher across all treatment groups compared with previous measurements (Figure 4).

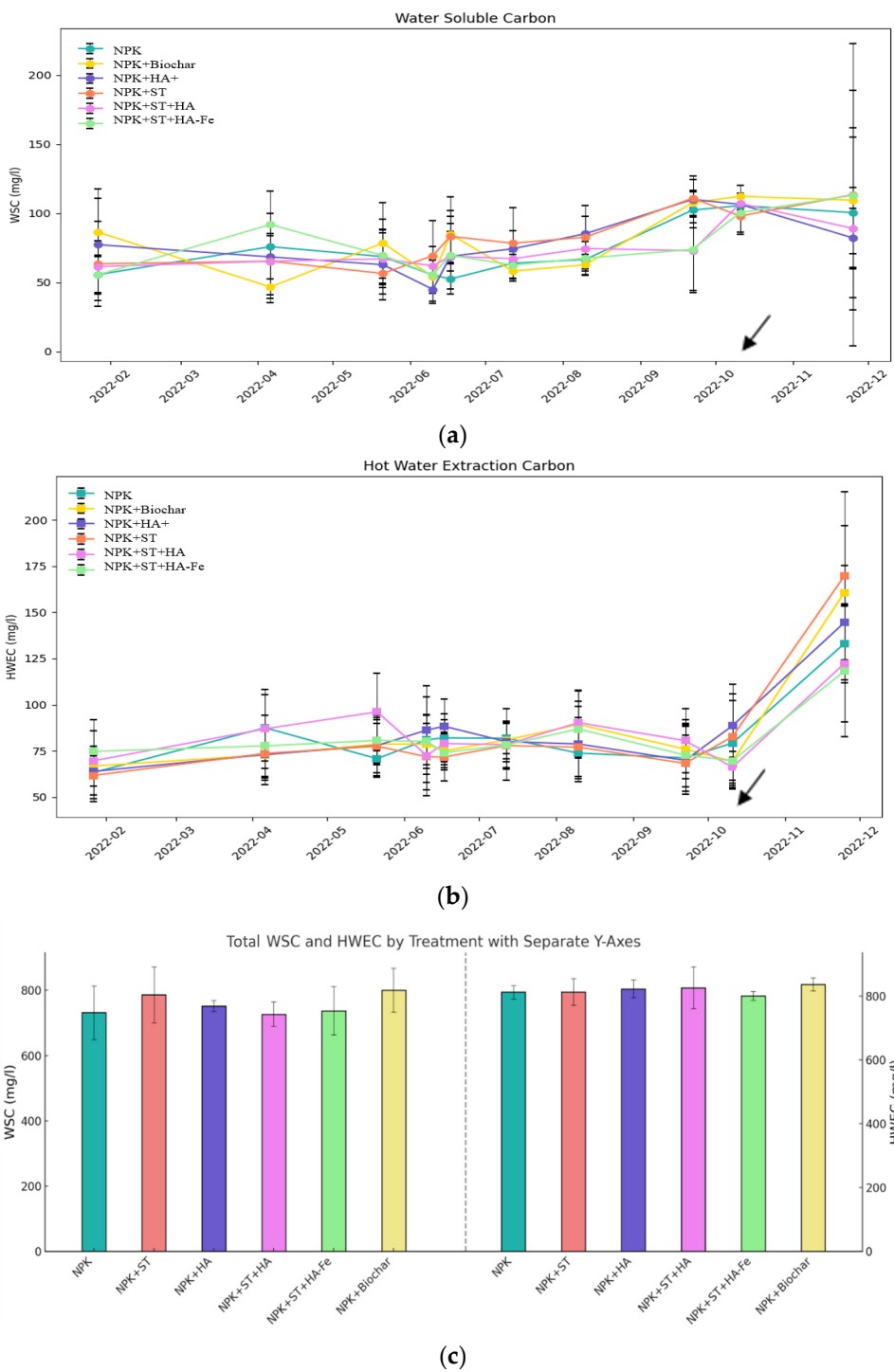

**Figure 4.** (**a**) Daily changes in WSC (↗: harvest day); (**b**) daily changes in HWEC (↗: harvest day); (**c**) total WSC and HWEC. Notes: NPK, application of N-$P_2O_5$-$K_2O$ fertilizer; ST, application of straw; HA, application of humic acid; HA-Fe, application of humic acid–iron complex; Biochar, application of biochar.

### 3.3. Greenhouse Gas Emissions

Regarding methane emissions, all treatment groups showed a gradual increase in $CH_4$ concentration following rice transplantation, with almost no emissions observed after complete drainage (DAT 106). Notably, the highest $CH_4$ concentration ($65.9 \pm 18.22$ mg m$^{-2}$ h$^{-1}$) was observed 23 days after transplantation in the rice straw-treated plots. In contrast, the NPK, HA, and Biochar treatment groups exhibited a similar trend with the highest $CH_4$ concentrations observed around 70 days post transplantation. NPK and HA-Fe treatments exhibited relatively lower $CH_4$ emissions compared with other rice straw-mixed treatments after 50 days of transplantation. The change in carbon dioxide emissions did not show any particular trend during the flooding period across all treatment groups. However, HA-Fe-treated plots maintained higher levels of $CO_2$ concentration than other treatments. After complete drainage for harvest (DAT 106), $CO_2$ emissions sharply increased in all treatment groups. Nitrous oxide emission changes increased during the application of basal dressing (DAT 15) and panicle fertilizer (DAT 70), with a sharp increase in $N_2O$ concentration in all treatment groups after the soil became reductive following complete drainage (Figure 5).

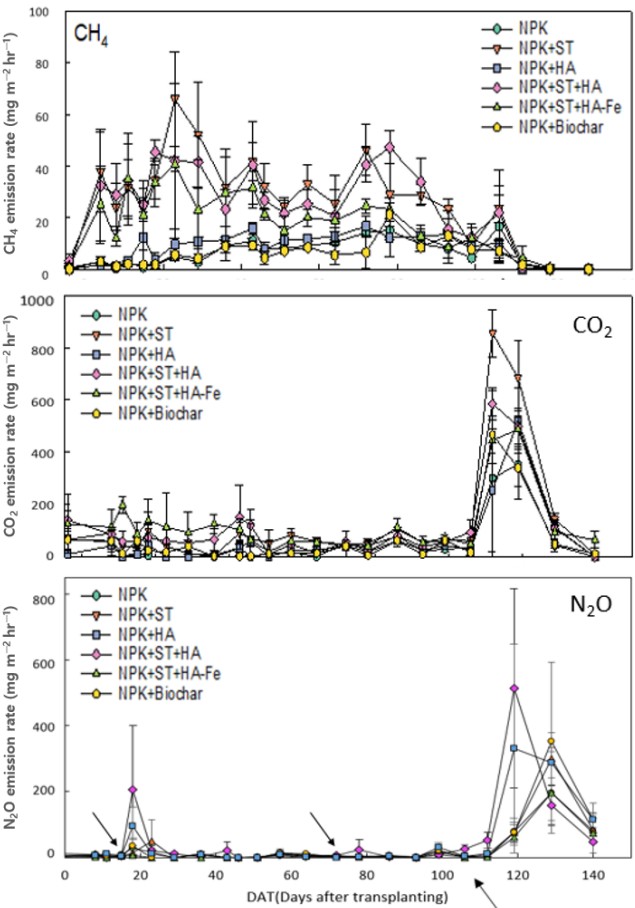

**Figure 5.** Daily changes in greenhouse gas ($CH_4$, $N_2O$ and $CO_2$) emission concentrations (↘: Fertilizer application ↖: Complete water drainage). Notes: NPK, application of N-P$_2$O$_5$-K$_2$O fertilizer; ST, application of straw; HA, application of humic acid; HA-Fe, application of humic acid–iron complex; Biochar, application of biochar.

The total quantitative analysis of $CH_4$ emissions revealed that the amount was highest in the following order: ST, ST+HA, ST+HA-Fe, HA, NPK, and Biochar. Methane emissions were significantly decreased (34.4%) when rice straw was treated with the HA-Fe compared with rice straw alone. Biochar treatment resulted in a 9% decrease compared with NPK treatment, but the difference was not statistically significant. A significant reduction

in methane was observed when the HA-Fe was added to rice straw treatments (34.4% decrease). Total $CO_2$ quantitative analysis indicated that emissions were highest in the following order: ST, ST+HA-Fe, ST+HA, HA, Biochar, and NPK. $CO_2$ emissions were higher in the treatments in which rice straw was applied (ST, ST+HA, ST+HA-Fe) compared with those without straw application (NPK, Biochar, HA). There was a statistically significant difference between groups treated with and without rice straw, but no difference within the groups. Total $N_2O$ quantitative analysis showed emissions in the following order: NPK, Biochar, HA, ST+HA, ST+HA-Fe, and ST, but no significant difference was observed between the treatment groups ($p = 0.105$) (Figure 6).

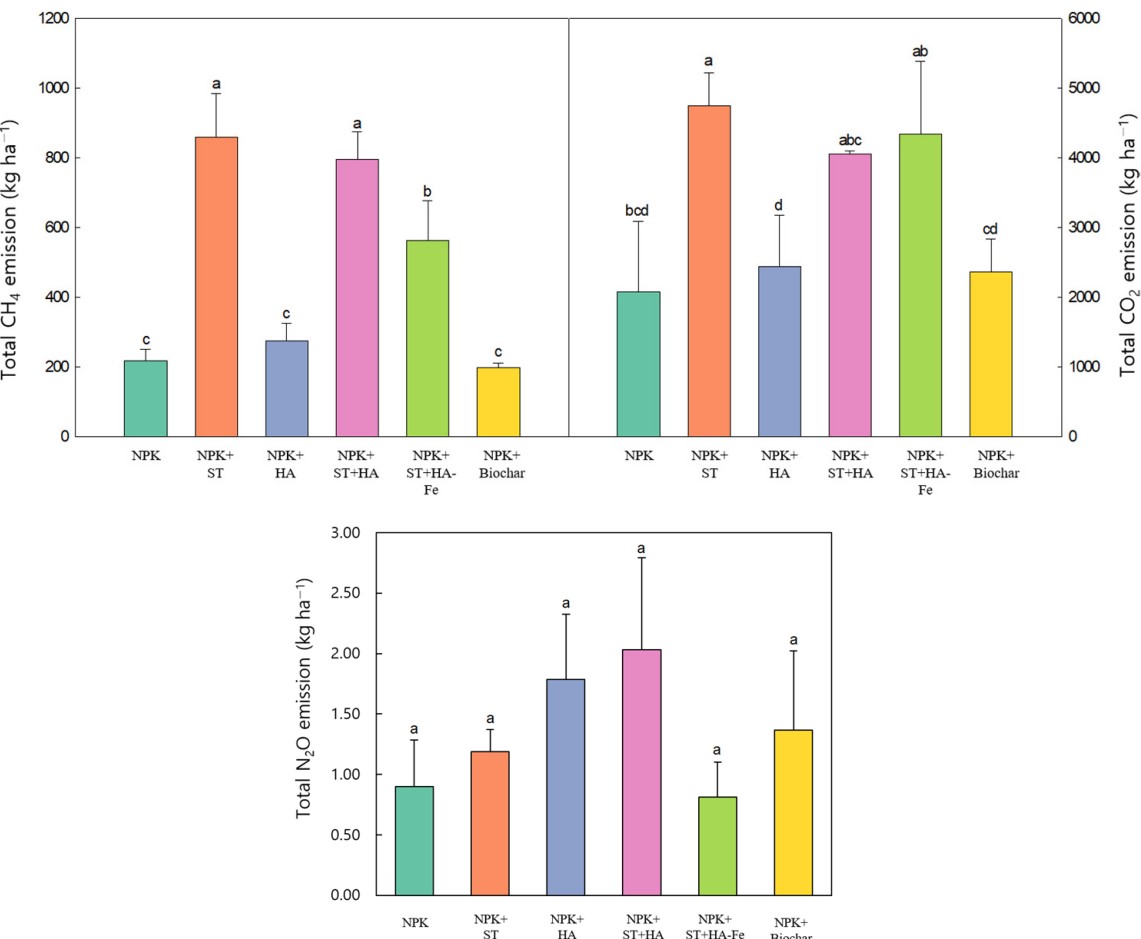

**Figure 6.** Total greenhouse gas ($CH_4$, $N_2O$ and $CO_2$) emissions during the growing season. Note: NPK, application of N-$P_2O_5$-$K_2O$ fertilizer; ST, application of straw; HA, application of humic acid; HA-Fe, application of humic acid–iron complex; Biochar, application of biochar; different letters following each value in the same column indicate significant difference at $p < 0.05$.

### 3.4. Rice Growth and Yield

The number of rice tillers was highest in the following order: the Biochar, HA, ST+HA, ST+HA-Fe, NPK, and ST treatment groups. Plant height was tallest in the following order: the ST+HA, Biochar, HA, ST, NPK, and ST+HA-Fe treatment groups. The grain and straw yield were relatively higher in the biochar-treated plots, but no statistically significant differences were observed between the treatment groups (Table 4).

**Table 4.** Growth properties and grain yield.

| Treatment | Plant Height (cm) | No. of Tillers per Plant | Grain Yield (kg ha$^{-1}$) | Straw Yield (kg ha$^{-1}$) |
|---|---|---|---|---|
| NPK | 94.6 ± 3.30 cd | 9.2 ± 2.02 c | 5284 ± 290.1 a | 5935 ± 1574.6 a |
| NPK+ST | 96.0 ± 3.37 bc | 8.6 ± 2.60 c | 5246 ± 282.4 a | 4869 ± 1340.9 a |
| NPK+HA | 97.9 ± 3.98 bc | 12.0 ± 1.61 a | 5258 ± 421.4 a | 5163 ± 936.8 a |
| NPK+ST+HA | 101.8 ± 4.02 a | 11.2 ± 1.88 ab | 5641 ± 470.5 a | 5074 ± 563.1 a |
| NPK+ST+HA-Fe | 91.4 ± 4.07 d | 9.5 ± 1.50 bc | 4791 ± 918.3 a | 4993 ± 1236.8 a |
| NPK+Biochar | 98.2 ± 3.43 b | 12.6 ± 3.05 a | 5664 ± 36.1 a | 7267 ± 389.1 a |

Notes: Mean ± SD (*n* = 3), NPK, application of N-P$_2$O$_5$-K$_2$O fertilizer; ST, application of straw; HA, application of humic acid; HA-Fe, application of humic acid–iron complex; Biochar, application of biochar; different letters following each value in the same column indicate significant differences at *p* < 0.05.

### 3.5. Net GWP and Yield-Scale GHGI

The Net GWP was calculated by converting the total emissions of methane and nitrous oxide to $CO_2$ equivalents using GWP indices and then adding these to the total $CO_2$ emissions. Net GWP among treatment groups was the highest in the following order: ST, ST+HA, ST+HA-Fe, NPK, HA, and Biochar. Consistent with the effects on methane emission reduction, the treatment groups with rice straw had significantly higher emissions compared with those without straw application. Adding the HA-Fe to rice straw resulted in a significant reduction in total emissions (33.9% decrease). Greenhouse gas intensity (GHGI, kg $CO_2$-eqv. kg grain$^{-1}$) was more than twice as high in the treatment groups with rice straw compared to those without, but no significant differences were observed within each group (Table 5).

**Table 5.** Seasonal fluxes of greenhouse gases, net global warming potential (GWP), and yield-scaled greenhouse gas intensity (GHGI).

| Treatment | Seasonal Flux (kg ha$^{-1}$) | | Net GWP (Mg CO$_2$-eqv. ha$^{-1}$) | Yield-Scaled GHGI (kg CO$_2$-eqv. kg grain$^{-1}$) |
|---|---|---|---|---|
| | CH$_4$ | N$_2$O | | |
| NPK | 217 ± 32.8 c | 0.9 ± 0.38 a | 5.1 ± 0.68 c | 0.9 ± 0.07 b |
| NPK+ST | 859 ± 126.4 a | 1.1 ± 0.18 a | 18.3 ± 2.74 a | 3.4 ± 0.36 a |
| NPK+HA | 276 ± 50.7 c | 1.7 ± 0.54 a | 6.2 ± 0.96 c | 1.1 ± 0.08 b |
| NPK+ST+HA | 796 ± 80.0 a | 2.0 ± 0.77 a | 17.0 ± 1.73 a | 3.0 ± 0.51 a |
| NPK+ST+HA-Fe | 563 ± 113.9 b | 0.8 ± 0.29 a | 12.0 ± 2.49 b | 2.6 ± 0.99 a |
| NPK+Biochar | 198 ± 13.1 c | 1.3 ± 0.66 a | 4.7 ± 0.34 c | 0.8 ± 0.06 b |

Notes: Mean ± SD (*n* = 3), NPK, application of N-P$_2$O$_5$-K$_2$O fertilizer; ST, application of straw; HA, application of humic acid; HA-Fe, application of humic acid–iron complex; Biochar, application of biochar; different letters following each value in the same column indicate significant difference at *p* < 0.05.

## 4. Discussion

### 4.1. The Effect of Rice Straw Application

The application of rice straw significantly increased methane emissions compared with NPK treatment (Figure 6). Generally, the introduction of organic matter, such as rice straw, into flooded paddies provides a substrate for anaerobic methanogenic bacteria, leading to methane production [11,35,36]. The rice straw treatment group (ST) showed a tendency for methane emissions to reach a peak approximately one month after transplantation. The application of rice straw (600 kg per 10 ares) resulted in a 2.59–3.95-fold increase in methane emissions compared with the untreated (NPK) group. These results suggest that rice straw acts as a source of carbon (C) and provides readily utilizable substrates for methanogenic bacteria, thus increasing methane emissions, which is consistent with the results of other studies [35,37–40]. Moreover, to understand the carbon consumption pathways of methanogenic bacteria, we analyzed easily microbial-usable organic carbon (HWEC, WSC) in the soil. The lack of significant differences in HWEC and WSC content among the treatment groups indicates that the source of carbon consumed by methanogenic bacteria is supplied through rice straw application rather than soil carbon (Figure 4).

Despite being organic matter, stable forms of organic materials such as composts of cow dung and leaves do not promote methane production; instead, they can actually reduce it [41,42]. However, because NPK was applied to all treatment groups, no positive effects of rice straw on growth or yield were observed. There were no significant differences in nitrous oxide emissions, which are typically released from paddy soils during the nitrogen cycle when the soil is in an oxidative state [43–45]. Because all treatment groups were managed under flooded conditions, we inferred that there was no difference in nitrous oxide emissions resulting from water management. When comparing the greenhouse gas intensity by converting methane and nitrous oxide emissions into $CO_2$ equivalents using the GWP and then dividing by grain yield, it was evident that the rice straw application groups significantly increased greenhouse gas emissions.

### 4.2. The Effect of Biochar Application

Our results indicate that applying biochar manufactured from rice husks to paddies reduced methane emissions by 8.75% compared to the single fertilizer (NPK) treatment; however, there was no significant difference in nitrous oxide emissions or other factors. This contrasts with a meta-analysis conducted on East Asia, which found that biochar can be used as a soil amendment to increase rice production and reduce methane emissions [13]. Although 168.8 kg of carbon per 10 ares was applied using the carbon content of the biochar, this did not directly lead to an increase in methane emissions, suggesting that biochar was not utilized as a substrate by methanogenic bacteria. The biochar's manufacturing conditions, with an H/C molar ratio below 0.7 and an O/C molar ratio below 0.4, ensure microbial decomposition stability [46,47]. The results from the analysis of soil labile carbon in Figure 4 suggest that the short-term impact of biochar on the soil was minimal over one year. In terms of rice growth and yield, biochar had no direct impact on yield, but significant differences were observed in height increase (3.8%) and tiller number (36.9%) compared to the single fertilizer (NPK) treatment. Biochar's extensive surface area and well-developed pore structure are reported to improve soil conditions, enhance moisture retention and microbial richness, thereby positively affecting plant productivity [48]. Long-term monitoring of biochar application has shown a positive effect on crop yield, with yield increases correlating positively with the amount of biochar applied and attributed to improvements in soil organic carbon (SOC) [49]. Our results from a single-year trial indicate that while biochar improved rice growth, it did not lead to increased yields or significantly contribute to SOC enhancement. In conclusion, when calculating the greenhouse gas intensity (GHGI), biochar appeared to be the most effective reduction measure compared to all treatments in reducing greenhouse gases. However, as there was no statistical difference from the NPK treatment in this one-year study, further multi-year research is deemed necessary to verify its effectiveness [13,46,47].

### 4.3. The Effect of HA and HA-Fe Application

HA, sourced from MYCSA, was applied at a rate of 15 kg per 10 ares. Compared with the untreated (NPK) group, the application of HA did not result in significant differences in methane and nitrous oxide emissions. In terms of rice growth, while plant height showed no difference, the tiller number increased by 30.4% compared with the untreated group, but there were no significant differences in yield (grain and straw). When rice straw and HA were applied together, methane emissions decreased by 7.3% compared with rice straw alone, although it was not statistically significant. Significant increases ($p < 0.05$) in plant height and tiller number were observed, but there were no differences in yield. HA is known to improve soil structure, stimulate microbial activity, and regulate soil pH, which can benefit crop production [50,51]. However, in this one-year study, aside from an increase in tiller number (30.4%), there was no reduction in greenhouse gas emissions or significant yield benefits compared with the untreated group. The combination of rice straw and HA resulted in significant increases in plant height (6%) and growth (30.2%) compared with straw alone. The HA-Fe was evaluated because it was anticipated to yield positive results

in greenhouse gas emissions, rice growth, and rice yield. Iron (III) ions, serving as electron acceptors, induce an oxidative state in paddy fields and inhibit methanogenic bacteria activity, thus reducing methane emissions [52]. When applied with rice straw, a 34.4% decrease in methane was observed compared with straw alone, although there was no significant difference in nitrous oxide emissions. The application of the HA-Fe resulted in a 4.79% decrease in plant height, but no significant differences in tiller number or yield. A significant reduction of 33.2% in Net GWP was observed compared with straw alone, and a 29.4% reduction compared with ST+HA treatment, indicating significant results. However, there were no significant differences in greenhouse gas intensity among the various straw applications. The methane reduction effect of the HA-Fe suggests that the inhibitory effect of iron ions on methanogenic bacteria [53], along with the refractory organic material, humic acid, acting as an electron acceptor in anaerobic microbial respiration [15], play an important role in suppressing methane emissions.

## 5. Conclusions

In this study, we conducted a one-year trial to evaluate the effects of humic acid, a HA-Fe, rice husk biochar, and rice straw on greenhouse gas emissions and crop yield in rice paddies in Korea. Treating paddies with both the HA-Fe and rice straw led to a 34.4% reduction in methane emissions without decreasing rice yield, compared to treatment with rice straw alone. Our findings suggest that the HA-Fe can be an effective means of reducing methane while preserving crop yield.

**Author Contributions:** Conceptualization, H.-S.G.; data curation, H.-S.L. and H.-S.G.; formal analysis, H.-S.L. and H.-S.G.; investigation, H.-S.L., H.-S.G. and S.-R.L.; methodology, H.-S.G.; project administration, H.-S.L.; supervision, T.-K.O.; validation, T.-K.O. and H.-S.L.; visualization, T.-K.O. and H.-S.L.; writing—original draft, T.-K.O. and H.-S.L.; writing—review and editing, T.-K.O., J.-M.L., D.-G.P., S.-I.L., S.-R.L., S.-H.E. and H.-R.P. All authors have read and agreed to the published version of the manuscript.

**Funding:** This work was carried out with the support of the Cooperative Research Program for Agriculture Science and Technology Development (RS-2021-RD009054), Rural Development Administration, Republic of Korea.

**Institutional Review Board Statement:** Not applicable.

**Informed Consent Statement:** Not applicable.

**Data Availability Statement:** The data presented in this study are available on request from the corresponding author.

**Conflicts of Interest:** The authors declare no conflicts of interest.

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
