# Peer review of "Reducing Methane Emissions with Humic Acid–Iron Complex in Rice Cultivation: Impact on Greenhouse Gas Emissions and Rice Yield"

_sustainability, doi:10.3390/su16104059_

Round 1
Reviewer 1 Report
Comments and Suggestions for Authors
1.In line 81, the Latin name (Oryza sativa L.) should appear in the first occurrence of rice, and the Latin name needs to be italicized.
2.The line charts of Figure 4 and 5 are better changed to histograms, which are currently stacked on top of each other, making it difficult to distinguish the differences between the groups, and the results of the significant analysis of the differences can be supplemented after changing to histograms.
3.The three legends in Figure 5 are of different sizes, please adjust them consistently.
4.Some data need to confirm the results of significant differences. In Figure 6, the standard deviations of the two sets of data in the total CH4 emission overlapped, and it is necessary to confirm whether there is a significant difference. The results of the significance difference analysis need to be supplemented in the total N2O emission.
Comments on the Quality of English LanguageSome words are not used properly, please check and revise them throughout.
Reviewer 2 Report
Comments and Suggestions for Authors
The regular paper of Lee et al. entitled: «Reducing Methane Emissions with Humic Acid-Iron Complex in Rice Cultivation: Impact on Greenhouse Gas Emissions and Rice Yield» exhibits different methods applied by authors to reduce greenhouse emission from rice paddy field due the microbial activity. The authors showed that the treatment using rice straw (ST) combined with Humic acid (HA) together with Iron (Fe) as a complex reduces significantly the methane emission by around 34.4%; however, when authors applied methods containing rice straw, the emission of methane was significantly increased compared to groups without any addition (NPK group). The beneficial effect, of these methods, is the fact that they do not compromise rice growth and productivity. This reveals that rice derivatives (Straw and Biochar) in paddy field favourites greenhouse gas emission. The paper is quite clean and well designed.
Authors applied different methods to sort out which one is more appropriate for alleviating greenhouse gas emission from paddy rice fields. Accordingly, they found that the application of straw treatment (ST) together with HA-Fe complex constitutes the best method to reduce methane emission. The method used to relieve methane emission from rice paddy fields is interesting and farmers are recommended to follow this method to decrease pollution and global warming, which may aggravate climate changes. The method described herein represents a low-cost method that all famers can apply, but with substantial positive repercussions on the environment management and air quality. It helps mitigating the effect of greenhouse gas on the life quality. However, regarding the methodology, the experiments designed meet the global goal requirement and I don’t see any additional expedients are needed or any other control is demanded.
The references listed in the work are all appropriate and fit well the manuscript content. Most of them talk about the greenhouse gas emission management. The conclusions also reflect the manuscript content. The authors replied on the main message addressed in the manuscript and they found that the presence of rice straw exacerbate the methane emission; however, the use of rice straw humic acid and iron decrease strongly the greenhouse emission. The data quality is quite good and data are consistent and work is well implemented. Figure quality is not that high but they can fit the journal level. It is mostly acceptable work, with an average novelty level for readers. English quality is ok but need to double check.
No revisions. it is quite clean manuscript
Reviewer 3 Report
Comments and Suggestions for Authors
The manuscript is well-design, but it misses the mechanism resulting lower gas emission although manuscript cited other papers
1. The title should be “Humic Acid-Iron Complex Reduced Methane Emissions and Improved Rice Yield”
2. Abstract:
a. The statement in lines 12-14 is not appropriate since this issue can appear all over the globe, not only Korea. Thus, this statement should be in the introduction not the abstract. Therefore, this statement in the abstract should be revised in a different way.
b. The doses of rice straw and humic acid should be more detailed.
c. “Treatment with rice straw (ST), rice straw + humic acid (ST+HA), rice straw + humic acid-iron complex (ST+HA-Fe), biochar (Biochar), and humic acid (HA) to a single fertilizer (NPK) control group” is too ambiguous.
d. The findings are sketchily presented in the abstract. The results should be more detailed.
e. The authors should explain the reason why rice straw rice straw + humic acid iron complex can reduce gas emision
3. Introduction:
a. As mentioned above, the phenomenon of greenhouse gases emission happens everywhere in the world, not only Korea. Please find more significant works on this around the world.
b. The introduction is not sufficient. It should introduce each aspect of the study, such as rice farming and greenhouse gases emission, methane emission and its harms, humic acid, humic acid-Fe complex, and biochar, etc. in each separate paragraph. It means that materials used in this study should introduce
c. Before the objective statement, the novelty and the necessity of the study should be declared.
d. “(HA)” line 58.
e. Methods should not appear in the introduction.
4. Materials and Methods:
a. What is the distance between the treatment plots?
b. “Oryza sativa” line 81.
c. “10a”? line 87, 90, 327.
d. Table 2, why is there only BET Surface Area that has SD value?
e. Typos and grammar errors can be found abundantly. Please check the manuscript again by a native speaker or service.
f. Please be uniform in describing unit. For example, only one can be used between kg-1, /kg, per kg. The same should be applied to other units.
g. Table 3, why are some parameters bold?
5. Results:
a. Citations are not allowed in the results.
b. Methods should not be repeated in this section either.
6. Discussion:
a. “Yu Jiang et al. (2019)” is not a proper citation.
b. The result is repeated too much in the discussion.
c. The section lacks comparisons between the results of the current study and the ones of the previous studies.
7. Conclusions:
a. In my opinion, only significant findings should be presented in the conclusion.
b. Explanations and discussions should be limited in the conclusion as well.
Comments on the Quality of English LanguageMinor editing of English language required
Round 2
Reviewer 3 Report
Comments and Suggestions for Authors
All my comments were addressed, the manuscript is significantly improved
Please do not mention again what the research perform, so remove the first sentence in lines 374-376; the phrase ‘the results showed that’ should delete
Comments on the Quality of English LanguageMinor editing of English language required
Author Response
Thanks for the comment. The sentence has been revised.